



# Improvements of a low-cost CO₂ commercial NDIR sensor for UAV atmospheric mapping applications

Yunsong Liu[1,2], Jean-Daniel Paris[1,2], Mihalis Vrekoussis[2,3], Panayiota Antoniou[2], Christos Constantinides[2], Maximilien Desservettaz[2], Christos Keleshis[2], Olivier Laurent[1], Andreas Leonidou[2], Carole Philippon[1], Panagiotis Vouterakos[,2], Pierre-Yves Quéhé[2], Philippe Bousquet[1], Jean Sciare[2]

[1]Laboratoire des Sciences du Climat et de l'Environnement, CEA-CNRS-UVSQ, UMR8212, IPSL, Gif sur Yvette, 91191, France
[2]Climate and Atmosphere Research Center (CARE-C), the Cyprus Institute, Nicosia, 2113, Cyprus
[3]Institute of Environmental Physics and Remote Sensing (IUP) & Center of Marine Environmental Sciences (MARUM), University of Bremen, Bremen, D-28359, Germany

*Correspondence to*: Yunsong Liu (yunsong.liu@lsce.ipsl.fr)

**Abstract.** Unmanned Aerial Vehicles (UAVs) provide a cost-effective way to fill in gaps between surface in-situ observations and remote-sensed data from space. In this study, a novel portable $CO_2$ measuring system suitable for operations on-board small-sized UAVs has been developed and validated. It is based on a low-cost commercial nondispersive near-infrared (NDIR) $CO_2$ sensor (Senseair AB, Sweden), with a total weight of 1058 g, including batteries. The system performs in situ measurements autonomously, allowing for its integration into various platforms. Accuracy and linearity tests in the lab showed that the precision remains within ±1 ppm (1σ) at 1 Hz. Corrections due to temperature and pressure changes were applied following environmental chamber experiments. The accuracy of the system in the field was validated against a reference instrument (Picarro, USA) onboard a piloted aircraft and it was found to be ±2 ppm (1σ) at 1 Hz and ±1 ppm (1σ) at 1 min. Due to its fast response, the system has the capacity to measure $CO_2$ mole fraction changes at 1 Hz, thus allowing the monitoring of $CO_2$ emission plumes and the characteristic of their spatial and temporal distribution. Details of the measurement system and field implementations are described to support future UAV platform applications for atmospheric trace gas measurements.

## 1 Introduction

According to the IPCC (2021), the global mean temperature will increase by at least 1.5 °C in the next 20 years relative to the pre-industrial period for all scenarios. This warming, attributed to human activities, is driven by the increased emissions of heat-trapping greenhouse gases (GHGs) in the atmosphere. Impacts of global warming, such as heatwaves, extreme precipitation events, sea-level rise and biodiversity loss are already visible, affecting human societies and natural ecosystems (IPCC 2018). Because of its importance, global warming has become one of the most critical challenges of the 21st century from both a scientific and societal perspective. To tackle global warming, almost all members of the United Nations agreed to join forces to keep the warming below 2 °C (ideally 1.5 °C) under the Paris Agreement of 2015. This agreement intensifies



the need to strengthen our capacity of having high-quality and accurate observations of atmospheric GHG at all scales including local, regional and global measurements both at the surface and vertically resolved. Atmospheric concentration measurements from various platforms can therefore be used to estimate emissions at different scales.

Carbon dioxide ($CO_2$) is the most abundant, human-released GHG, in the atmosphere. Notably, the $CO_2$ mole fraction recently reached a new high in 2020 of $413.2 \pm 0.2$ µmol mol$^{-1}$ (ppm), which is 149 % higher than its pre-industrial level (WMO, 2021). About 86 % of total $CO_2$ emissions emanate from fossil fuel combustion, with around 25 % of it being taken up by the oceans and 31% by land surfaces (Friedlingstein et al., 2021).

Systematic in-situ ground-based measurements of $CO_2$ started in 1958 in Mauna Loa in Hawaii (Pales and Keeling, 1965).
Since then, in-situ measurements at many locations but also from various mobile platforms (e.g., cars and ships) have significantly improved our knowledge of the $CO_2$ spatial and temporal distribution (Daube et al., 2002; Agustí-Panareda et al., 2014; Liu et al., 2018; Defratyka et al., 2021; Paris et al., 2021). Throughout time, in-situ measurements have been complemented by remote sensing providing space-based global observations of $CO_2$ column-averaged mole fraction data from various instruments (Bovensmann et al., 1999; Turner et al., 2015; Jacob et al., 2016; Wunch et al., 2017; Suto et al., 2021).
Meanwhile, $CO_2$ instrumentation onboard airborne platforms have been developed in the past 20 years (e.g. Watai et al., 2006; Sweeney et al., 2015). These measurements are meant to fill the gap between ground-based observations and remote sensing space-based observations to better represent $CO_2$ spatial distribution at large scales. However, manned (piloted) aircraft which can carry standard analyzers are costly and complex to organize, requiring frequent maintenance (Berman et al., 2012; Bara Emran et al., 2017). Furthermore, at smaller geographical scales (landscape, industrial assets, urban area), manned airborne
platforms have strong limitations and cannot fly at low speed in all areas. UAVs have been demonstrated to be useful to detect and map emission plumes of other trace gases because of their ability to operate at very low speed/altitude and with slow cruising speeds (e.g. Barchyn et al., 2017). Additionally, UAVs, unlike piloted aircraft, can operate over hazardous areas such as volcanic eruptions and forest wildfires. However, until now very few calibrated $CO_2$ measurements have been reported in the literature (Kunz et al., 2018) due to the challenge of measuring this species with sufficient precision.

A large part of the anthropogenic $CO_2$ originates from point emission sources such as power plants burning fossil fuels (Pinty et al., 2017; Reuter et al., 2021). An appropriate sensor for UAV platforms would have the potential to provide independent $CO_2$ measurements across these source plumes to verify mitigation strategies. Often the $CO_2$ signals of strong emitters can be mixed with strong biospheric signals even at local scales. In addition, the planetary boundary layer (PBL) dynamics can strongly influence atmospheric concentrations. It is therefore important to separate the influence of exogenic factors and isolate
the contribution from targeted emission plumes. Another potential application of a UAV-$CO_2$ system is to document the spatial distribution of $CO_2$ around fixed observations. Watai et al. (2006) argued that UAVs have the potential to provide measurements close to the surface and inside the PBL complementary of data obtained from fixed observatories such as tall towers, and make frequent and simultaneous measurements in multiple locations at low cost. In this case, UAV measurements help separate signal variability into a large-scale footprint of ground stations and variability due to local influences. Despite
these challenges, there have been ongoing efforts to develop compact, lightweight, and low-powered GHG sensors, able to be





integrated into UAVs to address these needs. Berman et al. (2012) developed a highly accurate UAV greenhouse gas system (but heavy: 19.5 kg) for measuring carbon dioxide ($CO_2$) and methane ($CH_4$) mole fraction. Malaver et al.(2015) integrated a non-dispersive infrared (NDIR) sensor (3285 g) for $CO_2$ measurement into a solar-powered UAV for effective 3D monitoring. Kunz et al. (2018) reported the development of a high accuracy ($\pm1.2$ ppm) $CO_2$ instrumentation well-suited for UAVs.

However, the commercial $CO_2$ sensor used in the study was disassembled and redesigned, making it difficult to replicate widely. Allen et al (2019) applied a UAV-$CO_2$ sensor system to infer a landfill gas plume. Chiba et al. (2019) developed a UAV system (2.7 kg) to measure regional $CO_2$ mole fraction and obtain vertical distributions within 1.75 ppm standard deviation over a farmland area and deduced vegetation sink distribution from their results. More recently, Reuter et al. (2021) developed a lightweight (about 1.2 kg) UAV system to quantify $CO_2$ emissions of point sources with a precision of 3 ppm at

0.5 Hz.

These works have faced the difficulty to miniaturize high-precision, fast-response $CO_2$ sensors. Few studies among them could reach a $CO_2$ measurement accuracy below 2 ppm with light payload (2 kg) on board UAVs. It is also challenging to have stable and high-frequency measurements against rapid changes in pressure and temperature, which is also the main reason for UAV-$CO_2$ measurements not being widely applied. Therefore, this study aims to develop a cost-effective, compact, lightweight $CO_2$

measurement system with high frequency and accuracy that can be widely used in different UAV applications. Targeted applications include emission estimates from point sources, stack emission factor measurements, as well as mapping $CO_2$ distribution in mixed natural/urban environments.

Towards this goal, a portable $CO_2$ sensor system has been developed based on a low-cost commercial NDIR $CO_2$ sensor (Senseair AB, Sweden). Prior to integration, the accuracy and linearity of the instrument were ensured with a series of

laboratory tests. The performance of the system was validated during laboratory (chamber) and ambient conditions. For the latter, the system was installed onboard a manned aircraft and unmanned aerial vehicle platforms. As a proof of concept, intensive flights of the developed UAV-$CO_2$ sensor system were presented in the urban area (Nicosia, Cyprus). It is shown that our system is easy to be reproduced, enabling a wide range of field applications, such as urban and point-source emissions monitoring. Moreover, the system developed in this study has the potential to accommodate other sensors to make stack

emission ratio measurements.

## 2 Methodology

### 2.1 $CO_2$ sensor

The sensor used in this study is a non-dispersive near-infrared (NDIR) sensor from SenseAir AB based on their High-Performance Platform (HPP) 3.2 version for sub-ppm gas detection. These sensors measure the molar fraction of $CO_2$ in the

optical cell based on IR light absorption, based on the Beer-Lambert (Barritault et al., 2013). The multi-pass cell of the sensor provides eight roundtrips of the beam with a total path length of 1.28 m. Temperature-controlled molded optics in the sensors are used to keep the temperature of the sensor cell constant to prevent condensation on the mirrors (Hummelgård et al., 2015).





This study involved two $CO_2$ sensor units using this technology (named SaA and SaB hereafter). More information on the sensor can be found in Arzoumanian et al. (2019).

**2.2 Laboratory tests**

The schematic diagram of the measurement setup used for laboratory testing is shown in Fig. 1. In this setup, the sampled air first passes through a 15 cm cartridge filled with magnesium perchlorate ($Mg(ClO_4)_2$), which is sufficient to dry air at a room temperature (24 °C) and a flow rate of 500 ml min$^{-1}$ to a water mole fraction of 20 ppm for 2 h; and then through a 0.5 μm membrane filter to remove particles. A diaphragm micro-pump (GardnerDenverThomas, USA, Model 1410VD/1.5/E/BLDC/12V) drives the air through the gas line towards SaA and SaB. Temperature and relative humidity are continuously monitored via a SHT75 sensor placed between the micro-pump and the two sensors. Finally, a Raspberry Pi3 acquires the data from all the sensors. The integrated system is powered by a 12 V DC supply, isolated from the UAV power system. Parallel to the two sensors, a Picarro model G2401instrument (Picarro, USA) based on cavity ring-down spectroscopy (CRDS) (Crosson, 2008) served as a reference instrument in this setup (see Figure 1).

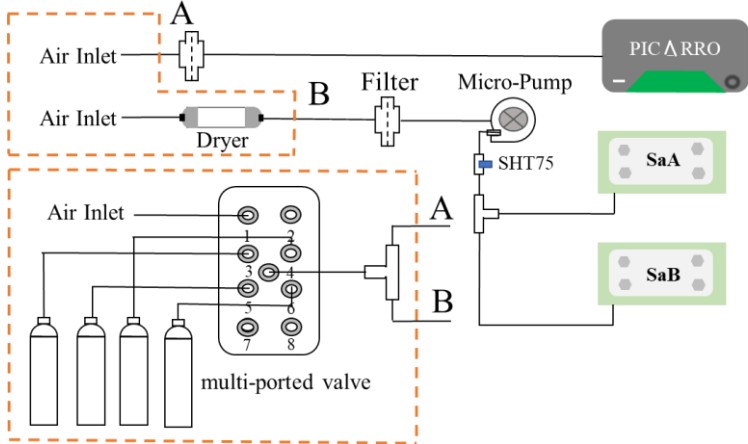

**Figure 1: The schematic of the developed system for lab tests and field deployment (A and B represent air flows to G2401 and $CO_2$ sensors, respectively).**

Figure 2 presents the data quality control procedure flow-chart. SaA and SaB were firstly tested in the metrology laboratory of the Integrated Carbon Observation System (ICOS) Atmosphere thematic center (ICOS ATC). Then the system was integrated into a manned aircraft and UAVs to be validated and evaluated under ambient conditions. Table 1 is a summary of all the laboratory and field tests performed for the system, and all results are presented in section 3. In the laboratory, four calibration sequences were performed to determine the calibration function that linked the measured values to the assigned values (Yver Kwok et al., 2015). Four high-pressure calibration standard gas cylinders with known amounts of $CO_2$, ranging from 380.096 ppm to 459.773 ppm, were used. The standard gases were calibrated using the international primary standard



for GHG, maintained in NOAA CMDL, Boulder, Colorado, USA (www.esrl.noaa.gov/gmd/ccl/). To ensure stabilization after adequate flushing of each sensor's cell with $CO_2$, each standard gas ran for 30 min continuously and only the last 10 min of data were used. Then the calibration function using a linear fit was calculated for the sensors and the Picarro instrument. The cylinder with 459.773 ppm $CO_2$ was considered to resemble ambient atmospheric conditions. During the Allan Deviation test

(Hummelga˚rd et al., 2015), the $CO_2$ sensors continuously measured a cylinder filled with dry air for 24 h.

Temperature (T) and pressure (P) sensitivity tests were performed in a closed automated climate chamber at the Observatoire de Versailles Saint-Quentin-en-Yvelines (OVSQ) Guyancourt, France, using the Plateforme d'Integration et de Tests (PIT). The temperature (from -60 °C to 100 °C) and pressure (from 10 hPa to 1000 hPa) ranges inside the chamber can be controlled and supervised by the Spirale 2 software (https://www.ovsq.uvsq.fr/essais-thermiques). We implemented repeated sequences

of variable temperature and pressure following (Arzoumanian et al., 2019). These tests allow determining the linear response of SaA and SaB sensors against temperature and pressure (as shown in Section 3.2).

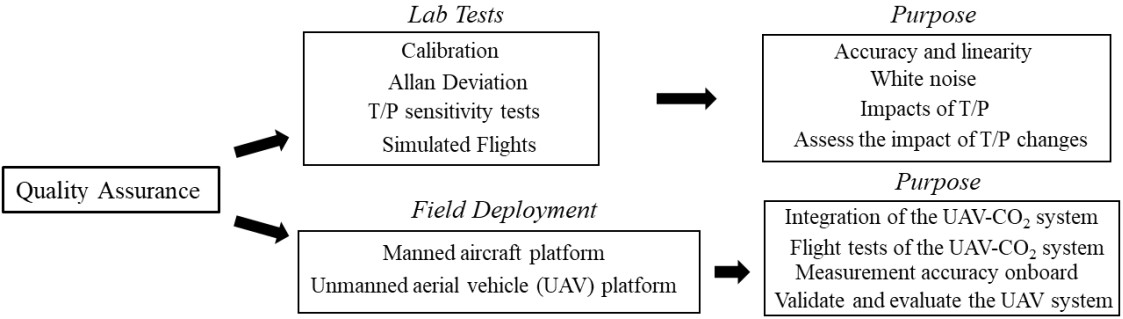

**Figure 2: The flow chart of data quality control procedures.**

**Table 1: A summary of the laboratory tests and field deployment.**

| Code name | Name | Purpose | Parameters | Range of T (°C) and P (mbar) | Duration (h) | Sensors tested |
|---|---|---|---|---|---|---|
| CA | Calibration | Test calibration frequency and stability | $CO_2$ | N/A | 8 | SaA, SaB |
| AD | Allan Deviation | Illustrate white noise, stability and detection limit | $CO_2$ | N/A | 24 | SaA, SaB |
| PT1 | Temperature and pressure sensitivity tests | Correlation between $CO_2$ and P/T | T, P | 0-45 and 600-1000 | 72 | SaA, SaB |
| PT2 | Temperature and pressure sensitivity tests | Correlation between $CO_2$ and P/T | T, P | 0-45 and 600-1000 | 72 | SaB |





| SF1 | Simulation flight | Estimate P/T impact | T, P | 15-25 and 800-1000 | 4 | SaB |
| SF2 | Simulation flight | Estimate P/T impact | T, P | 15-35 and 600-1000 | 10 | SaB |
| Aircraft Test | Manned aircraft Test | Estimate precision onboard | T, P, $CO_2$ | Real flight conditions | 2.5 | SaA, SaB |
| UAV Test | UAV Test | Test and evaluate the system on UAVs | T, P, $CO_2$ | Real flight conditions | ~0.3 | SaB |


## 2.3 Aircraft test

After a series of laboratory tests, the sensors were moved to a manned aircraft together with a reference instrument Picarro G2401-m to test the performance of SaA and SaB under real atmospheric conditions.

SaA, SaB and a reference Picarro instrument G2401-m were flown onboard a manned aircraft on April 8, 2019 in the vicinity

of Orleans forest (150 km south of Paris), France. All instruments were calibrated using standard cylinders from ICOS-ATC before and after the flight (Hazan et al., 2016). The setup used and the aircraft are shown in S1. These flights aimed to confirm the accuracy of SaA and SaB in real flight conditions.

## 2.4 Unmanned Aerial Vehicle (UAV) system integration

Then, for further validation the system was minutarized and integrated into a small-size Unmanned Aerial System (UAS),

developed at the Unmanned Systems Research Laboratory (USRL) of the Cyprus Institute (CyI) (https://usrl.cyi.ac.cy/). The components of the integrated system are shown in Fig. 3a. The $CO_2$ sensor setup weighs 1058 g with dimensions of 15 cm × 9.5 cm × 11 cm, including the battery. A 15 cm customized cartridge replaced here to reduce volume and weight. The impact of water vapor dilution on dry $CO_2$ mole fraction is within 40 ppb by using the dryer. It does not depend on external systems, allowing for its integration into various small UAVs. The system was successfully integrated into the USRL small-sized quad-

rotor UAS (Fig. 3b), optimally developed in terms of minimum size and maximum performance, to accomplish the desired $CO_2$ unmanned measurements. Multi-rotors allow vertical take-off and landing (VTOL) in urban and remote regions (Kezoudi et al., 2021). The UAS has up to 30 min flight endurance for atmospheric measurements with the selected sensor. In order to improve accuracy and response time for in-fight temperature measurements (critical for $CO_2$ correction), a Rotronic HC2-ROPCB sensor (Rotronic, Switzerland) replaced the SHT75 sensor. To validate the system on site, calibration sequences were

performed before and after the flights in the laboratory. In addition, a target gas cylinder was performed for 20 min between each flight to determine and correct the instrument's drift over time.



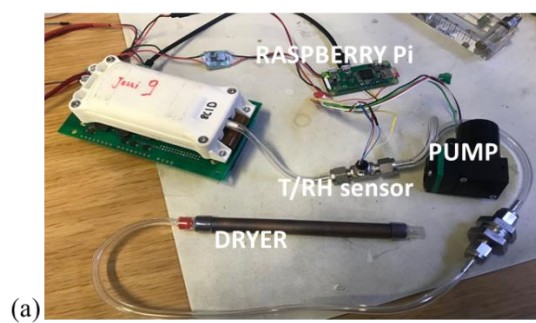 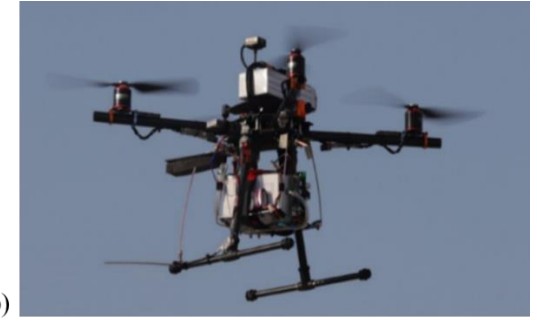

**Figure 3: Components of the portable CO$_2$ sensor system setup (a) and the selected UAV (b).**

## 3 Results

### 3.1 Sensor calibration

The response curves obtained from the CO$_2$ calibration are shown in S2. The stability of successive CO$_2$ calibrations is shown in Fig. 4, which presents the difference between CO$_2$ mole fraction measured by sensors and CO$_2$ mole fraction assigned to each calibration cylinder. The biases of SaA and SaB against the four calibration standards are negative and positive, respectively, during the calibration (Fig. 4). Additionally, the biases increased by 0.2 ppm on average between calibration sequences (2 h of each sequence). This drift against sensors' running time is further investigated and validated in the field deployment (Section 3.4). The result of the Allan Deviation (AD) test is shown in S3. The plot shows the stability as a function of integration time (Hummelgård et al., 2015). The unfiltered data were used from HPP data set. The precision improved by increasing the integration time. However, the sensors were intended for mobile platforms, their performance at 1 Hz was chosen as the most significant. The precision is respectively $\pm$ 0.36 ppm (1$\sigma$) and $\pm$ 0.85 ppm (1$\sigma$) for SaA and SaB at 1 Hz (S3), which shows the precision of the sensors in the laboratory is below 1 ppm (1$\sigma$) at 1 Hz.

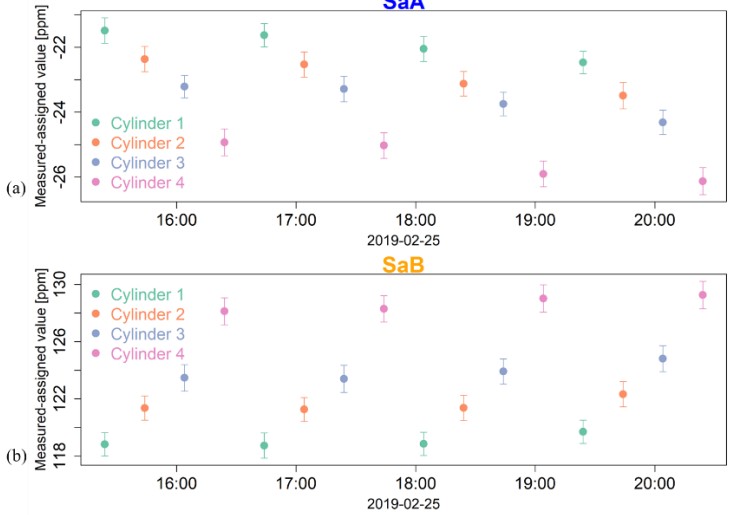



**Figure 4: Stability of successive $CO_2$ calibrations for SaA (a) and SaB (b), the error bars represent the standard deviation of 2-second averages.**

### 3.2 Temperature and pressure dependence

### 3.2.1 Temperature sensitivity test

During temperature sensitivity tests, the chamber pressure was kept constant at 950 hPa, while the temperature was gradually changed, as seen in S4. The temperature ranged between 0 °C and 45 °C, following 9 °C increment steps, lasting for 20 min. The sensors' cell temperature exhibited an unstable behavior for chamber temperatures below 25 °C, while it was stable, at approximate 57 °C, for chamber temperatures above 25 °C. However, SaA and SaB behaved oppositely when their cells'

temperature changed. Therefore, two scenarios were considered for both sensors.

The first scenario is when the analyzer's cell temperature is stable while the ambient air temperature changes (above 25 °C). The trend coefficients of $CO_2$ mole fraction over ambient temperatures were -0.564 and -0.527 for SaA and SaB, respectively (shown in Fig. 5a and Fig. 5c). The second scenario is when both the analyzer's cell and ambient temperatures change simultaneously. In this case, the impact of ambient air temperature changes obtained from the first scenario has been corrected

prior to considering the cell temperature changes. The  trend coefficients of $CO_2$ mole fraction over cell temperatures were -0.979 and 0.378 for SaA and SaB, respectively (shown in Fig.5 b and d). Consequently, SaA performed better when applying the temperature sensitivity test (high $R^2$, lower standard error).

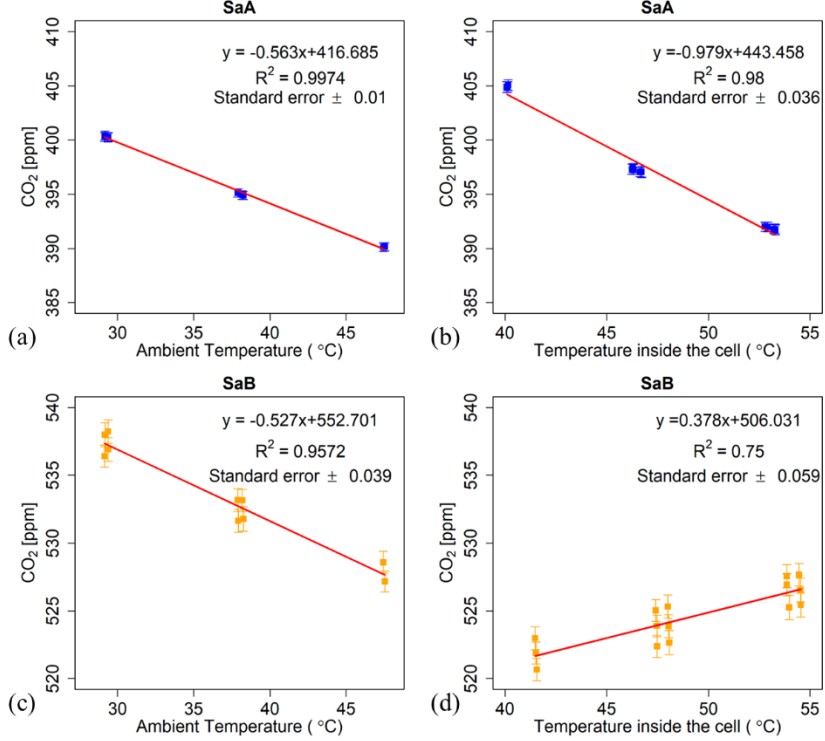





**Figure 5: Temperature sensitivity tests in the environment chamber, (a) and (c) represent the first scenario, (b) and (d) represent the second scenario.**

### 3.2.2 Pressure sensitivity test

During the pressure tests, the chamber temperature was maintained at 25 ℃, and pressure ranged from 600 hPa corresponding to 3 km above sea level (ASL) to 1000 hPa in 100 hPa steps, repeated twice. SaA and SaB performed significantly differently in this test, with the SaB sensor showing increased sensitivity to pressure changes (Fig. 6). Generally, the sensors have an internal pressure correction from the manufacturer and it is not implemented to SaB apparently. However, SaB performed better in the pressure sensitivity test, with tighter linearity (higher $R^2$) when both tests were accounted for.

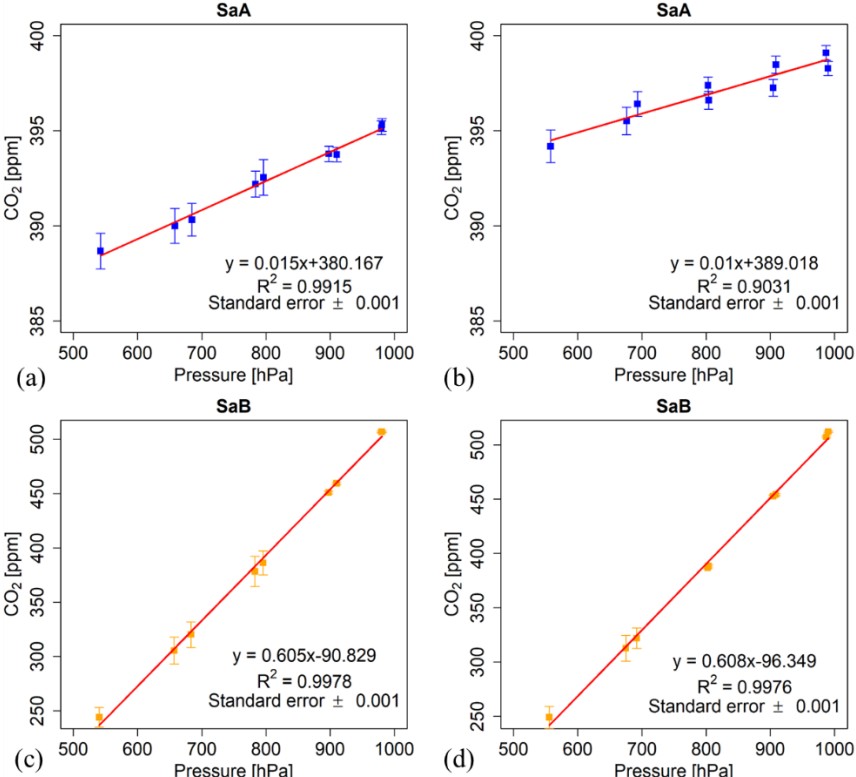

**Figure 6: Pressure sensitivity tests in the environment chamber, (a) and (b) represent SaA results of the repeated pressure tests, (c) and (d) represent SaB results of the repeated pressure tests.**

From the sensitivity tests presented above, we derived the following equations for both sensors:

SaA: $C_{cor}=C_{obs}+0.564\times(Ta-Ta_0)+0.979\times(Tc-Tc_0)-0.013\times(P-P_0)$       (Equation 1)

SaB: $C_{cor}=C_{obs}+0.527\times(Ta-Ta_0)-0.378\times(Tc-Tc_0)-0.607\times(P-P_0)$       (Equation 2)

Where $C_{cor}$ is the mole fraction after corrected for P/T changes. $C_{obs}$ is the observed mole fraction. Tc represents the analyzer's measurement cell temperature and $Tc_0$ is the original cell temperature at the start of the measurements. Ta represents the





ambient temperature and $Ta_0$ is the ambient temperature at the start of the measurement. P represents the ambient pressure and $P_0$ is the ambient pressure at the start of the measurements. The equations are also applied for calibrations.

Replications of temperature and pressure sensitivity tests for SaB at a later stage showed high consistency with the initial results presented above. Both sensors have shown different responses in the tests. Therefore, we recommend to characterize, at least once, the pressure and temperature sensitivity characterizations before field applications.

## 3.3 Manned aircraft test results

SaA and SaB measured consistently with the Picarro G2401-m for atmospheric pressure above 800 hPa (equal to 1.5 km ASL) (see Fig. 7a). Their precision was ±1.4 ppm (1σ) and ±1.7 ppm (1σ) at 1 Hz, 0.78 ppm (1σ) and ±1.1 ppm (1σ) with minute averaged data respectively (Fig. 7b), larger than the precisions calculated during the laboratory tests. This degradation was expected due to less optimal measurement conditions. Therefore, the test on the piloted aircraft shows sensors' precision onboard under the real flight condition is within 2 ppm (1σ) at 1 Hz and improves to about 1 ppm (1σ) with minute averaged data.

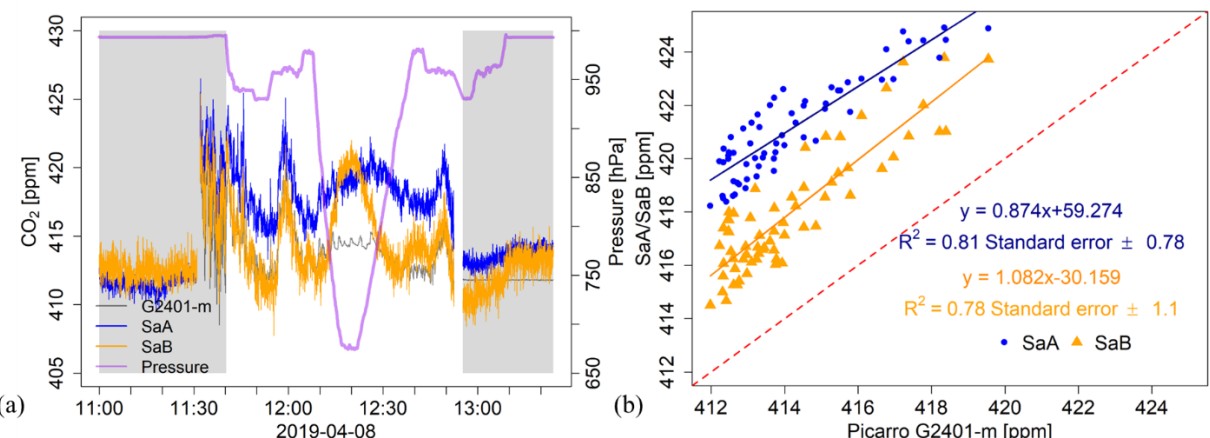

**Figure 7: Manned aircraft results, (a) is the time series and the grey shaded parts present measurements on the ground and measurements of the gas cylinder; (b) is the correlation between CO₂ sensors and G2401-m.**

## 3.4 Unmanned Aerial Vehicle (UAV) tests and validation

SaB was chosen for field deployments due to technical issues with SaA. SaB was integrated into a quad-rotor to evaluate and validate the performance of the sensor onboard a UAV platform during flights. The flight path was over the Athalassa National Forest Park (35.1294° N, 33.3916° E) in Nicosia, Cyprus (Fig. 8). Four flights were performed on June 10, 2021 from 1500 to 1800 LT. The procedure was the following: calibration response curves were obtained before and after the flights. A target gas cylinder was measured for 20 min between each flight to characterize the instrument drift. The sensitivity correction Eq. (2) was then applied to the raw data. It was noted that the measured target gas mole fraction drifted linearly throughout the day (S5a). To account for that, a time-dependent correction, based on running time, was calculated and applied for calibration



sequences (S5a). Practically, this correction was applied to obtain flight-specific calibration response curves according to the sensor running time and confirmed by the target linear drift (S5).

Reference $CO_2$ measurements were additionally conducted with another Picarro G2401 on the roof of the Novel Technologies Building (NTL) at the Cyprus Institute (CyI) (Fig.8), at 174 m ASL, 1.82 km northwest upwind from the UAV launching location (187 m ASL). Therefore, the flight path was downwind from the Picarro G2401. The residual values of $CO_2$ between Picarro and UAV-$CO_2$ systems varied from 0.2 ppm to 2.1 ppm (median =1.1 ppm) during the experiment.

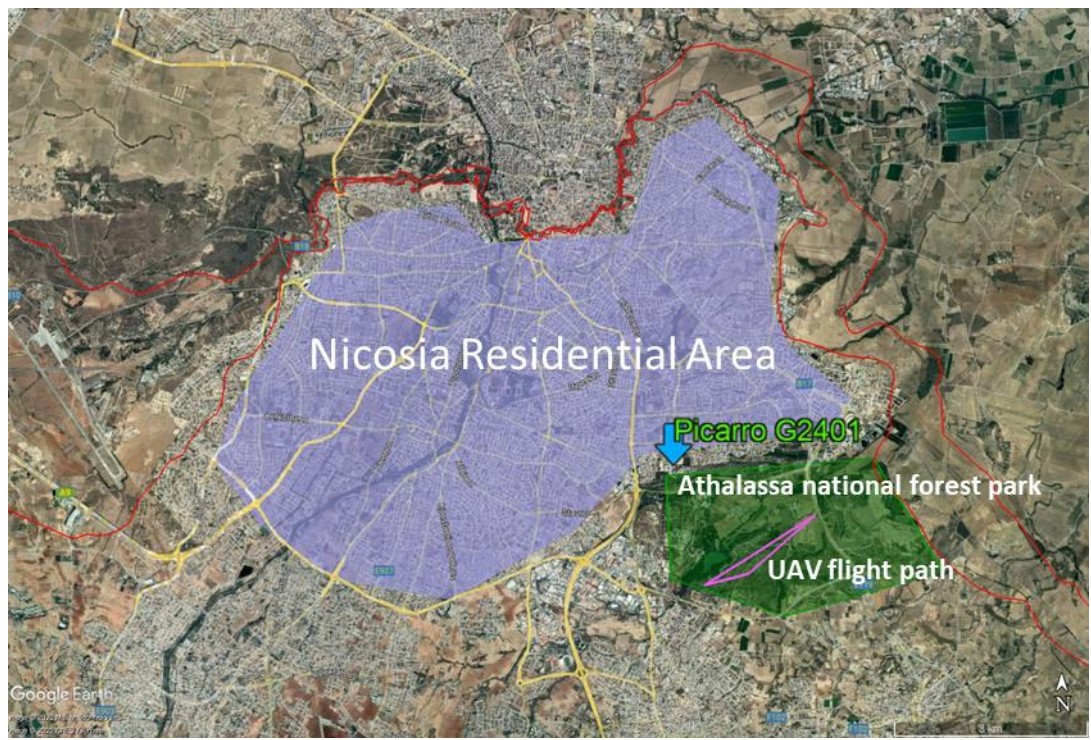

**Figure 8: The map presents the locations of the Picarro G2401 at CyI, the UAV flight path, Athalassa National Forest Park and the residential area in Nicosia (© Google Earth 2022).**

**4 Case study for $CO_2$ measurements in urban environment (Nicosia)**

The field campaign to test operation in real conditions of our UAV $CO_2$ system was performed on May 14, 2021 from early morning 0600 (LT) to late afternoon 1730 (LT). It took place above the Athalassa National Forest Park located southeast of

CyI in Nicosia, where 16 flights were performed. Each flight lasted approximately 15 min with most of the flight performed at a constant altitude of 50 m and 100 m above ground level (AGL) alternatively. The altitudes were determined following security rules. Firstly, the UAV had to maintain a safe distance above the treeline of the forest park. Therefore, the lowest safe altitude to fly the drone was 50 m AGL. Secondly, the ceiling of the UAV-$CO_2$ flights was decided to 100 m AGL, following the European regulations (2019/947 and 2019/945) for UAV operations in sparsely populated areas (open category A2), with

flights permitted up to 120 m AGL. The two selected altitudes were used alternatively in order to obtain representative





measurements for either horizontal "mapping" or vertical gradients. The vertical gradients were completed at lower altitudes by rooftop measurements in a nearby building. $CO_2$ mole fractions, as well as meteorological conditions, were measured during the flights on the roof of NTL at CyI. $CO_2$ measurements were done using a Picarro G2401 (174 m ASL, 16 m AGL, 35.141° N, 33.381° E); wind speed and wind direction were measured using a sonic anemometer Clima Sensor US model 4.920x.x0.00x
with a resolution of wind speed 0.1 m s$^{-1}$ and wind direction 1 °.

Each pair of 50 m and 100 m altitude flights lasted approximately 1 h (including flight time and the time needed to change the dryer and battery on the ground). The 15 cm cartridge filled with magnesium perchlorate ($Mg(ClO_4)_2$) was changed every two flights. The first six flights (three pairs) were performed continuously from 0600 to 0900 (LT), as well as the last six flights from 1500 to 1730 (LT). In between, four flights (two pairs) took place between 1000 and 1100 (LT) and between 1300 and
1400 (LT).

According to the meteorological station data, the wind direction in the morning (before 0800 LT) was from the northwest, with an average wind speed of 1.2 m s$^{-1}$. Then the wind direction shifted to northeast and southeast during the day before 1300 LT, with an average wind speed of 0.9 m s$^{-1}$. Afterwards, the wind shifted back to northwest, but with stronger wind speeds (average of 5.3 m s$^{-1}$).

Figure 9a displays the measured $CO_2$ (ppm) time series from all UAV flights and the Picarro. The $CO_2$ mole fraction measured during the flights in the early morning and evening, when northwesterlies occurred, was consistent with that measured by the G2401. A $CO_2$ enhancement linked to morning traffic peak (from 0700 to 0800 LT) was detected at all altitudes. Interestingly, the two measurements eventually differed at 1000 LT, creating a vertical gradient: the $CO_2$ mole fraction measured onboard the UAV remained constant, whereas a decrease of about 5 ppm was measured by the G2401 on the ground.

During the day, with the surface wind direction shifting starting 0800 LT from northwest to northeast and then southeast, the Picarro G2401 progressively sampled air from the Athalassa National Forest Park. The park, with a total area of 8.4 km$^2$, is an oasis of greenery with many trees, shrubs and grasses located on the southeastern edge of Nicosia. Considering that the inlet of the G2401 is at the same altitude above sea level as the UAV launching location, the lower observed $CO_2$ mole fraction by the G2401 can most likely be attributed to the Athalassa National Forest Park acting as a surface sink taking up $CO_2$. The
reduction of traffic after peak hour can also play a role in the first part of the day, when the air was blowing from the north. At 50 m or 100 m height, the constancy of $CO_2$ mole fractions during the day can suggest a different origin for the air sampled depending on the wind direction at these altitudes (wind was not measured onboard the UAV). Potential origins may include "regional" air moving above the surface layer or a plume of emissions from the city lofted at a few tens of meters with a stratified airmass above the park.

During the afternoon, the progressive convergence of surface and UAV observations, with a decrease of UAV-$CO_2$ values, suggest either a diffusion of the surface signals in altitude or an enhanced atmospheric mixing. This explanation could be supported using an anemometer integrated onboard the UAV to provide additional wind data at various heights. Embarked wind measurements would have to be considered for future applications.





A $CO_2$ mapping during the traffic peak hour is shown in Fig. 9c combined with the flight path at 100 m (the red dot represents
the launching site). Figure 9b shows the corresponding $CO_2$ time series combined with wind direction (arrow head) and wind
speed (arrow length) information. The high mole fraction (20 ppm above background levels) probably originated from local
traffic emissions from the main road south-west of the Athalassa National Forest Park (Fig. 8). This finding highlights the
capability of the developed UAV-$CO_2$ sensor system to detect fast mole fraction changes and the potential to provide useful
insight into $CO_2$ emissions close to the ground in urban areas.

From the vertical profiles (Fig. 10), the difference between the 0600 and 0700 LT profiles highlights the traffic peak hour.
Additionally, we observed an increasing difference (about 3 ppm) between ground level and 50 m AGL, followed by a
difference (about 0.5 ppm) between 50 m and 100 m AGL from 0800 to 1300 LT when the air mass came from the Athalassa
National Forest Park with an average wind speed of 0.9 m s$^{-1}$. This suggests that the $CO_2$ mole fraction measured by the G2401
and UAV-$CO_2$ system represents local $CO_2$ characteristics and that the Athalassa National Forest Park acted as a $CO_2$ sink.

Later on, between 1500 and 1700 LT when the average wind speed increased (5.3 m s$^{-1}$), the $CO_2$ mole fraction at 50 m AGL
and 100 m AGL converged towards surface values. This suggests that the observed wind speed enhancement enabled a better
mixing of surface signals in altitude. However, the transport of well-mixed regional background air masses at the measurement
area could also be an alternative explanation (background $CO_2$ mole fraction is 418.9 ppm). Although we demonstrated the
usefulness of UAV measurements to capture horizontal and vertical $CO_2$ gradients in the planetary boundary layer in an urban
or periurban environment, a definitive explanation of this particular observation would be beyond the scope of this paper.

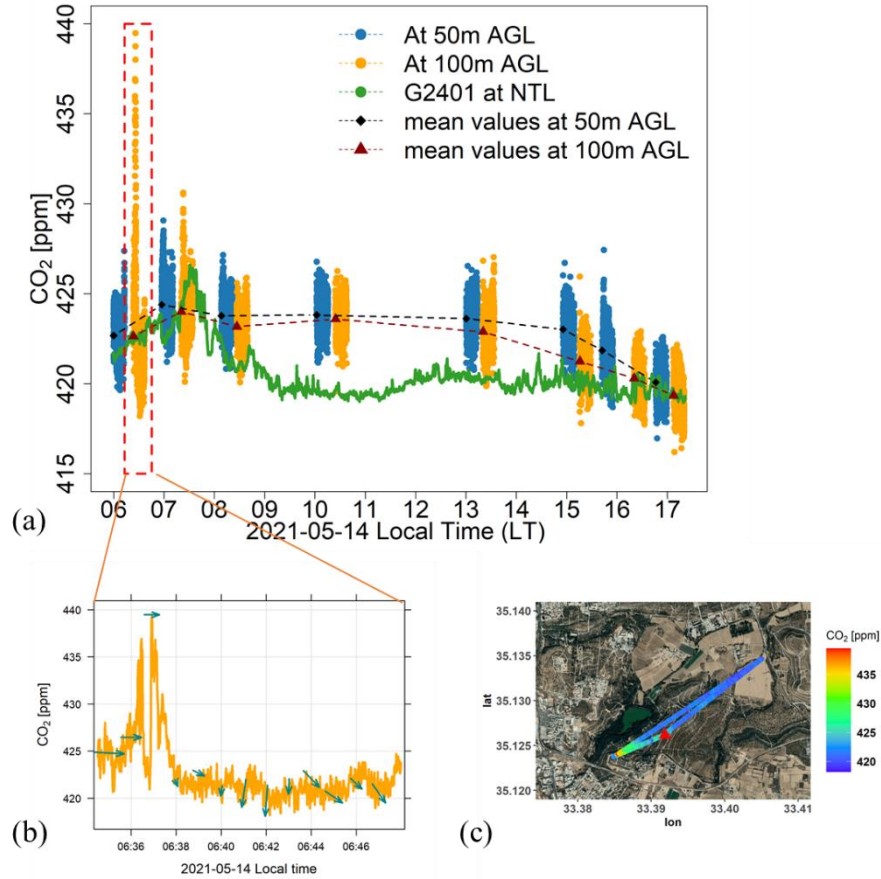

**Figure 9: (a) time series of CO₂ mole fraction measured by the UAV CO₂ sensor (at 50 m in blue and 100 m AGL in orange) and by the Picarro G2401 at CyI (in green). The black dots represent the averaged CO₂ mole fraction measured by SaB during the flights at 50 m, and the dark red dots represent the averaged CO₂ mole fraction measured by SaB during the flight at 100 m. (b) the corresponding CO₂ time series combined with wind direction (arrow head) and wind speed (arrow length) information obtained from the nearby meteorological station, which is a zoom of the second flight marked in the red dashed box in (a). (c) presents the CO₂ mapping (the red dot represents the launching location) during the rush hour (Map data: © Google, Maxar Technologies).**




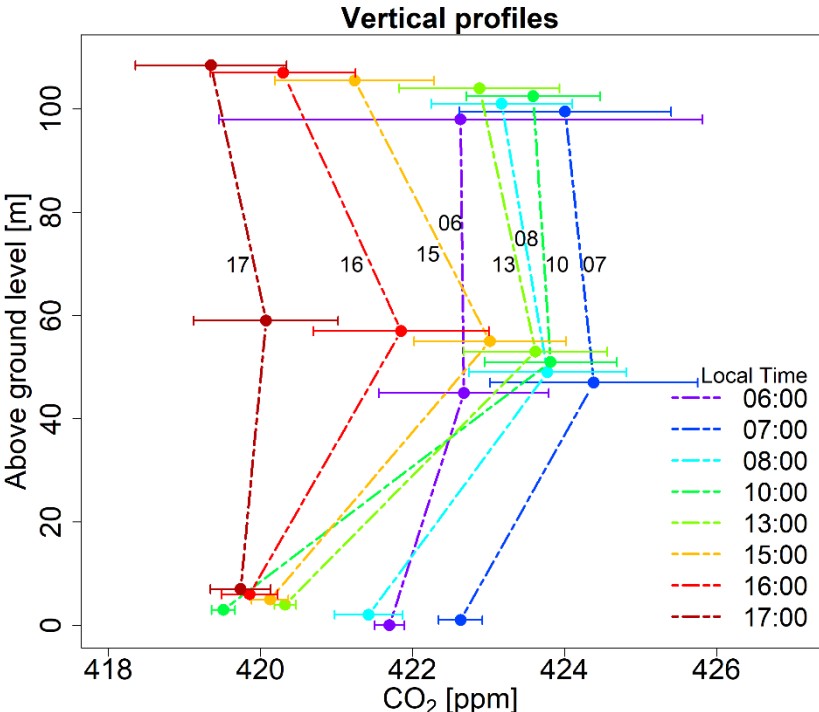

**Figure 10: Vertical profiles from the eight paris of flights. The ground level values are from the Picarro G2401 at CyI. CO$_2$ at 50 m**
**and 100 m AGL are from the UAV-CO$_2$ sensor horizontal flights, the error bars represent the standard deviation of the duration of each flight.**

## 5 Conclusions

Following the integration of a NDIR CO$_2$ sensor, we developed and validated an autonomous system that can be regarded as a portable package (1058 g), suitable for CO$_2$ measurements on board small UAVs (or other platforms) with good field

performance after applying calibration and data corrections (±1 ppm accuracy for 1 min averages). Prior to deployment, and in order to acquire high-quality observations, the sensor followed a series of quality control procedures. The laboratory tests indicated that the precision was within ±1 ppm (1σ) at 1 Hz. Two CO$_2$ sensors (SaA and SaB) were tested. Each sensor's performance is impacted by changes in pressure and temperature; therefore, it is necessary to perform pressure and temperature sensitivity tests before any field applications.

Further validation onboard a manned aircraft resulted in an estimated precision of ± 2 ppm (1σ) at 1 Hz and ±1 ppm (1σ) at 1 min time resolution. During the integration of our system onboard a small quad-copter, the calibration strategy has been extended to account for running-time-dependent instrumental drifts. Due to its simplicity, the developed system can be replicated easily for wider applications since it has compact, cost-effective and lightweight advantages. It is anticipated that the integrated portable package can be used in the investigation of emission ratios and fluxes, especially when combined with

other sensors on board the UAV platform.



As a proof-of-concept, the developed system had been deployed in a UAV-based flight campaign, where several horizontal flights were performed near the ground and up to 100 m in height. Mole fraction of $CO_2$ up to 440 ppm (20 ppm above the background levels) was detected during the morning traffic rush hour, attributed to emission from a major road located on the southwest of the Athalassa National Forest Park. The $CO_2$ mole fraction measured by the UAV system was consistent with

that measured by the Picarro G2401 at CyI when the flight path was downwind of CyI. The system also revealed its ability to capture the temporal variability of the vertical $CO_2$ gradient between the surface and the lower atmosphere. The observed $CO_2$ profiles depict the contribution of traffic emission in the morning from 0600 to 0800 LT, and also a probable sink due to the Athalassa National Forest Park during the course of the day from 0800 to 1300 LT. Furthermore, the measurement system captured the mole fraction drop from 1500 to 1700 LT observed at different height levels due to the intensification in the wind

speed leading to more horizontal and vertical mixing. In conclusion, the designed system demonstrated its capability to measure fast mole fraction changes and spatial gradients, and to provide accurate plume dispersion maps. It proved to be a good complementary measurement tool to the in-situ observations performed at the surface.

*Data availability.* The data presented in this study are based on many different experiments and given the fact that our experiments and field deployments were aimed at characterizing the two sensors used here. The data is not made publicly

available in a repository, but can be requested from the corresponding author.

*Author Contributions.* YL contributed to lab experiments' design and setup, field deployments, data collection, data analysis and writing of the manuscript. JDP, MV, PB and JS contributed to project advising, reviewing, and editing the manuscript. PA, CC, CK, AL, PV contributed to the miniaturization, integration and evaluation of the UAV-$CO_2$ sensor system, UAS operations and data collection. OL and CP contributed to laboratory experiments' setup, design and advising. PYQ contributed

to manned aircraft flight measurements, ground Picarro G2401 and meteorological station measurements and maintenance, data collection and providing supports during UAV-$CO_2$ flights. MD contributed to the review and editing of the manuscript.

*Competing Interests.* The authors declare that they have no conflict of interest

*Acknowledgements.* We would like to thank ICOS teams at LSCE for supporting a series of laboratory tests and aircraft field deployment flights. We are grateful for all the efforts of the technicians, pilots and software developers from USRL/CARE-C

at CyI for their support in the integration and field validation of the UAV-$CO_2$ sensor system.

*Financial support.* This research has been supported by the project Eastern Mediterranean Middle East-Climate & Atmosphere Research Center (EMME-CARE) which has received the funding from the European Union's Horizon 2020 research and



innovation programme under grant agreement No. 856612 and the Cyprus Government, and the project Air Quality Services
for clean air in Cyprus (AQ-SERVE) INTEGRATED/0916/0016 is co-financed by the European Regional Development Fund
and the Republic of Cyprus through the Research and Innovation Foundation.

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
