# Peer review of "Improvements of a low-cost CO2 commercial NDIR sensor for UAV atmospheric mapping applications"

_Atmospheric Measurement Techniques, 2022_

## Author Response (AR2)

Dear Prof. Grant Allen,

Thank you very much for your helpful review.

This study presents the development and validation of a novel portable $CO_2$ measuring system suitable for operations onboard small-sized UAVs. This system has a fast response time (1 Hz) and a relatively high precision ($\pm 2$ ppm $1\sigma$ at 1 Hz) to make it have the capacity to monitor emission plumes, and characterize their spatial and temporal distribution. Our revision following the two reviewers' comments tends to reinforce our statements about the importance of careful tests and calibrations to obtain measurements of sufficient precision.

Please find my detailed reply for each comment below.

**Specific comments and Questions:**

1/ Figure 4 and section 3.1 – the mean bias with respect to the calibration cylinders is large (but clearly measurable and correctable). As the two sensors tested in this work have significant (and different) positive and negative mean biases, this demonstrates that any individual sensor will always need a robust calibration (like the approach in the paper) to obtain meaningful data. This is an important statement to make clear in the paper to guide future users – i.e. that no SenseAir NDIR CO2 system should considered plug-and-play without conducting calibrations and bias correction prior to any measurement campaign and that data would suffer from extreme unknown biases without that important step. It may be worthwhile discussing that operational calibrations against any high precision instrument (e.g. Picarro or LGR or similar) may still be suitable for this task so long as those reference sensors are calibrated to NOAA/WMO gas cylinders themselves, i.e. a transferrable standard. This might avoid the step of future users needing to obtain expensive gas cylinders if they have access to other high precision reference instruments for example. It might be worth adding this to the discussion.

In the revised version, we will add the following sentences to Line 312, "It is essential to conduct calibrations before any measurements as shown in this study. NDIR $CO_2$ sensors should not be considered plug-and-play without conducting calibrations and bias correction prior to any measurement campaigns as measurement data would suffer from large, unknown biases without that important step. In general, we advocate that low- and mid-cost sensor units should systematically be characterized for their dependence to pressure and temperature, and their factory correction and calibration verified. Strategies for field deployment should also take into account the significant drift that can be observed at the hourly scale. Using a single target gas between flights is sufficient to cope

with this drift. Alternative strategies to correct the drift without using gas cylinders on the field remain to be explored, such as comparison against a high precision instrument at regular intervals during the deployment."

2/ Section 3.2. This is a very robust and rigorous calibration of the two specific sensors in this study and their (correctable) response to T and P, which look linear and encouraging. But, as rightly said in the paper, the equations derived are only applicable to these two specific sensors (and different for each sensor). A P and T calibration would need to be produced for any new unit, as suggested. This should be made extremely clear in the paper so that future users do not use the P and T relationships given in Equations 1 and 2, which only apply to these two specific units. Any information that could be given to help readers on this might be useful, e.g. might SenseAir provide those P and T calibrations with any new instrument or would those tests be something that users would always need to perform with a new sensor themselves? I see that it is recommended that users perform a T and P characterisation (line 208), so this mostly addresses the comment above, but it isn't clear if the guidance is to always perform this characterisation prior to any new measurement project, or it just needs to be done once for each unit (and therefore repeatable), or if this could this change over time, needing repeated characterisation?

In the revised version, we will replace the sentence at Line 208 with the following sentence: "Therefore, it is essential to perform both temperature and pressure sensitivity tests for individual sensors to obtain their individual correction equations against temperature and pressure changes. Here, we highly recommend to characterize every individual sensor at least once before any use. We also recommend to repeat regularly (e.g. annually) these tests as sensor performances tend to change over time".

3/ Line 53 - In the introduction section, it is claimed that "until now very few calibrated CO2 measurements have been reported in the literature" from aircraft. This is not accurate. Many papers from groups in the US, EU, and UK have reported calibrated CO2 measurements from instruments onboard manned aircraft over the past 10-15 years. For example, here is just a small sample of papers from a UK team, which describe the use of calibrated CO2 instrumentation (and their calibration procedures): Barker et al., 2020 - https://acp.copernicus.org/articles/20/15443/2020/ ; Pitt et al., 2018 - https://acp.copernicus.org/preprints/acp-2018-1033/acp-2018-1033.pdf; O'Shea et al, 204 - doi:10.5194/acp-14-13159-2014, 2014. I would recommend rephrasing this sentence to state that there are actually many high-precision calibrated aircraft CO2 instruments (citing a couple of papers like the ones above, or others), but that those instruments are typically very heavy and expensive and not suitable for UAV use.

We will add the following sentences to Line 53.
"Actually, high-precision calibrated $CO_2$ instruments have been deployed in manned aircrafts (e.g. Paris et al., 2008; Xueref-Remy et al., 2011; O'Shea et al., 2014; Pitt et al., 2018; Barker et al., 2020), but they are too heavy, large and expensive for UAV applications."

4/ Introduction – this is mostly a very thorough description of the state of the UAV CO2 field but it is missing some description of other current very high precision UAV CO2 sensors, e.g. the LGR Hoverguard system and AERIS sensors, which are a few kg in mass and now flown on UAVs – e.g. Shah et al., 2020 - https://amt.copernicus.org/articles/13/1467/2020/. But again, while those instruments are higher precision, I believe they may also be very expensive compared to the NDIR here, and also somewhat heavier. So much like comment #3 above, a short summary of higher-precision CO2 instruments (<0.1 ppm 1.s.d @ 1 HZ) and their pros and cons for UAV use compared with the NDIR, would be a very valuable addition to the introductory discussion.

After checking the report provided by LGR and the test report provided by ICOS-ATC on LGR pMGGA, the precision for $CO_2$ is below 0.2 ppm 1σ at 1 Hz. Therefore, the following sentences will be added to Line 75 in the revised version. "Moreover, very high-precision and commercial sensors (<0.2 ppm 1σ at 1 Hz) for UAV applications are emerging currently such as the ABB light Micro-portable Greenhouse Gas Analyzer (pMGGA) (Shah et al., 2020). However, the weight (about 3 kg) is much larger and the price is more expensive compared to the NDIR sensors mentioned in the above literature."

**Technical Corrections:**

1/ Remove space between number and percentage (there are no spaces between number and "%", only SI units)

The space will be removed in the revised version.

2/ Line 95 – add "law" to "Beer Lambert…."

 "Law" will be added in the revised version.

3/ Line 277 – "Embarked" may be better replaced with "UAV-integrated", or "installed".

 "Embarked" will be replaced by " UAV-integrated" in the revised version.

We kindly thank Anonymous Referee #2 for your helpful review.

This study presents the development and validation of a novel portable $CO_2$ measuring system suitable for operations onboard small-sized UAVs. This system has a fast response time (1 Hz) and a relatively high precision (±2 ppm 1σ at 1 Hz) to make it have the capacity to monitor emission plumes, and characterize their spatial and temporal distribution. Our revision following the two reviewers' comments tends to reinforce our statements about the importance of careful tests and calibrations to obtain measurements of sufficient precision.

Please find the detailed reply to each comment below.

P2L43: add ground-based remote sensing observations to the list. TCCON – Wunch et al.,2011 and COCCON – Frey et al., 2019.

We will add ground-based remote sensing observations and references in the revised version.

P1L25: please expand IPCC, add a reference to the report of 2021 and 2018 (line29)

We will add a reference "Khangaonkar et al., 2019" to line 29.

P3L73: the reference of Reuter et al. (2021) is not listed

The reference Reuter et al. (2021) will be listed in the revised version.

P15L312: -314: this is a particularly important message; perhaps the authors can put more emphasis on this in the main section of the paper and suggest some recommendations for future users.

The sentences "Therefore, it is essential to perform both temperature and pressure sensitivity tests for individual sensors to obtain their individual correction equations against temperature and pressure changes. Here, we highly recommend to characterize every individual sensor at least once before any use. We also recommend to repeat regularly (e.g. annually) these tests as sensor performances tend to change over time" will be added in the main section (line212-line216) in the revised version.